# Automatic Pruning Discovery for Large Language Models

Haidong Kang [1 2]   Lihong Lin [1]   Enneng Yang [3]   Hong-Ning Dai [4]   Hao Wang [5]

## Abstract

Large language models (LLMs) have achieved remarkable performance on a wide range of tasks, hindering real-world deployment due to their massive size. Existing pruning methods (e.g., Wanda) tailored for LLMs rely heavily on manual design pruning algorithms, thereby leading to *huge labor costs* and *requires expert knowledge*. Furthermore, we are the first to identify the serious *outlier value issue* behind dramatic performance degradation under high pruning ratios that are caused by uniform sparsity, raising an additional concern about how to design adaptive pruning sparsity ideal for LLMs. Can LLMs prune by themselves? In this work, we introduce an affirmative answer by proposing a novel pruning method called **AutoPrune**, which first overcomes expert knowledge limits by leveraging LLMs to design optimal pruning algorithm for themselves automatically without any expert knowledge. Specifically, to mitigate the black-box nature of LLMs, we propose a Graph-driven Chain-of-Thought (GCoT) to optimize prompts, significantly enhancing the reasoning process in learning the pruning algorithm and enabling us to generate pruning algorithms with superior performance and interpretability in the next generation. Finally, grounded in insights of outlier value issue, we introduce Skew-aware Dynamic Sparsity Allocation (SDSA) to overcome the outlier value issue, mitigating performance degradation under high pruning ratios. We conduct extensive experiments on mainstream LLMs benchmarks, demonstrating the superiority of AutoPrune, which consistently excels state-of-the-art competitors.

[1]Northeastern University, Shenyang, China [2]Hebei Key Laboratory of Marine Perception Network and Data Processing, Northeastern University at Qinhuangdao 066004, Hebei Province, China [3]Sun Yat-sen University, Shenzhen, China [4]Hong Kong Baptist University, Hongkong, China [5]Xidian University, Xian, China. Correspondence to: Haidong Kang <kanghaidong@qhd.neu.edu.cn>, Hao Wang <Haow@ieee.org>.

*Proceedings of the $43^{rd}$ International Conference on Machine Learning*, Seoul, South Korea. PMLR 306, 2026. Copyright 2026 by the author(s).

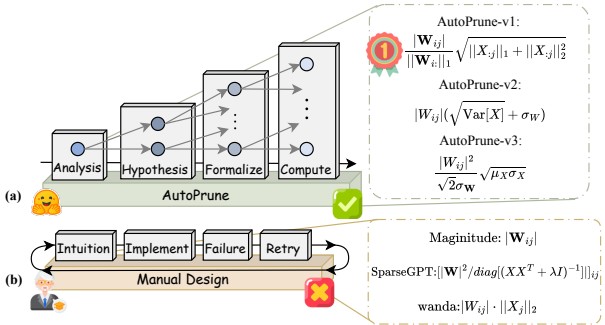

*Figure 1.* (a) AutoPrune *vs.* (b) Manual Design. Manual design requires expert knowledge, resulting in huge labor costs. In contrast, our AutoPrune can efficiently design several specialized pruning algorithms by leveraging LLMs.

## 1. Introduction

Large Language Models (LLMs) (Cha, 2022; Guo et al., 2025; Hurst et al., 2024; Touvron et al., 2023a) have attracted substantial attention from both academia and industry, which has reshaped the study of diverse tasks due to superior performance, e.g., language translation (Bubeck et al., 2023), code generation (Lai et al., 2025). However, the huge size and computational costs of LLMs (e.g., the GPT-175B (Brown et al., 2020) with 175 billion parameters, taking 350GB of memory with FP16 for inference) have become a key bottleneck for widespread real-world deployment in both edge and cloud scenarios (Zheng et al., 2025). This highlights the need to craft lightweight LLMs, facilitating deployment on resource-constrained devices.

So far, several pruning methods tailored for LLMs have been proposed (Frantar & Alistarh, 2023; Sun et al., 2024; Xie et al., 2024), liberating the LLMs from the above bottleneck in a training-free manner from the lens of unstructured pruning. In short, the core idea is to measure weight importance using statistical or theoretical properties of model weights or activations. For instance, Magnitude (Lee et al., 2021) is a superior pruning method for DNNs, which is also adopted for LLMs pruning due to only relying on the magnitude of weights. However, SparseGPT (Frantar & Alistarh, 2023) finds that "Magnitude" will result in dramatic performance degradation even under low levels of sparsity. To mitigate this problem, SparseGPT (Frantar &

Code is available at: https://github.com/yohbii/AutoPrune

Alistarh, 2023) adopts the Optimal Brain Surgeon (OBS) (Hassibi et al., 1993) to evaluate weight importance based on the statistical characteristics of the Hessian matrix, enabling selective pruning of insignificant weights. Moreover, SparseGPT suffers from huge computational budgets due to the expensive second-order optimization. To address this, Wanda (Sun et al., 2024) simplifies the computations by measuring the product of weight and activation magnitudes.

**Challenges.** Although these methods have taken the first step tailored for lightweight LLMs in an unstructured pruning way, they still suffer from critical limitations: (1) *rely heavily on extensive human expert knowledge through tens of thousands of trial-and-error (as shown in Fig. 1(b)), resulting in significant labor costs; (2) sensitivity to outlier weights, causing severe performance degradation under high pruning ratios (as depicted in Fig. 3 ).* Those issues highlight the need to rethink the design of pruning algorithms tailored for LLMs.

**Our New Observation.** Inspired by the advanced capacities of LLMs (e.g., code/text generation, contextual reasoning), we raise a new question: **Can LLMs prune by themselves?** In this paper, we propose a novel paradigm to automatically design pruning algorithms via LLMs (as depicted in Fig. 1(a)), which revolutionizes the design paradigm of pruning tailored for LLMs. Compared with previous methods crafted by human experts, our method can effectively and efficiently design pruning algorithms with optimal performance (as shown in Fig. 2) without any expert knowledge. Furthermore, we observe a fatal outlier value issue in LLMs by skewness (Seglen, 1992) (as shown in Fig. 3), which is sensitive to the pruning ratios of specific layers and dramatically decreases the performance of pruning methods. To reveal the root cause, we conduct an in-depth analysis of the underlying mechanism of the outlier value issue.

**Contributions.** In short, this paper aims to understand and address the aforementioned critical limitations caused by relying on extensive human expert knowledge and the outlier value issue. To tackle the limitations of relying on extensive human expert knowledge, we first revisit the mechanism of LLMs pruning (as shown in Table 2), and find that previous methods (i.e., SparseGPT, Wanda) are heuristics, relying on statistical or theoretical properties of weights or activations in LLMs. Motivated by the powerful knowledge generation capabilities of LLMs, we propose a novel pruning method (dubbed **AutoPrune**), which first revolutionizes the paradigm of LLMs pruning via automatically designing pruning algorithms by itself (as represented in Fig. 1). Before introducing our reasoning module, we first conduct a controlled study to understand whether *reasoning at all* helps an LLM design better pruning rules. We compare three variants under the same evaluation protocol: (i) a naive single-shot prompting baseline (no CoT), (ii) a linear step-

by-step CoT that follows one reasoning trajectory, and (iii) a graph-driven variant that explores multiple trajectories. As summarized in Table 1, the naive baseline performs worst (MMLU (Hendrycks et al., 2021) 27.25, WikiText-2 (Merity et al., 2016) 16.55), a linear CoT improves but remains limited (29.15, 7.17), while the graph-driven variant attains the best results (29.69, 6.28). These observations indicate that merely appending a single chain-of-thought is insufficient; exploring and selecting among multiple reasoning paths is crucial for robustly discovering stronger pruning rules. Notably, to obtain positive feedback of LLMs-driven pruning algorithms designed for the target task, we propose Graph-driven Chain-of-Thought (GCoT) to enhance reasoning by measuring positive gain from the target task, enabling LLMs to craft better pruning algorithms in the next generation.

To tackle the outlier value issue, we perform an experimentation by skewness analysis on LLaMA-1-7B with Magnitude (Fig. 3), validating the statement that existing methods suffer from the outlier value issue, some layers in LLMs are sensitive to the outlier value issue, which can lead to significant performance degradation under high pruning ratios. To reveal the root cause, we conduct an in-depth analysis of the outlier value issue (as shown in Fig. 4), and observe that uniform sparsity could be the potential reason for the outlier value issue, where the importance of the layer is ignored. Grounded in empirical observations via skewness analysis, we propose Skew-aware Dynamic Sparsity Allocation (SDSA), which perfectly overcomes the outlier value issue by preventing the over-pruning of certain layers and the over-preservation of others, leading to a more balanced and stable adaptation trajectory.

To sum up, the main contributions of this paper are:

- **New LLMs Pruning Paradigm.** To the best of our knowledge, this paper is the pioneering effort to propose a novel LLM-pruning paradigm by leveraging LLMs to design optimal pruning algorithms for themselves automatically without any expert knowledge, breaking expert knowledge limits.

- **Outlier Value Issue.** We thoroughly scrutinize the mechanisms of LLMs pruning through which high pruning ratios affect the performance, and first find that the outlier value issue in LLMs pruning. To reveal the root cause, we conduct an in-depth analysis and find that the outlier value issue is attributed to uniform sparsity.

- **GCoT driven Self-pruning**. To address the concern of black-box LLMs, we propose a GCoT to build a reasoning engine, shaping self-pruning by obtaining positive reward signals from target tasks.

- **Numerical Verification.** Extensive experiments show

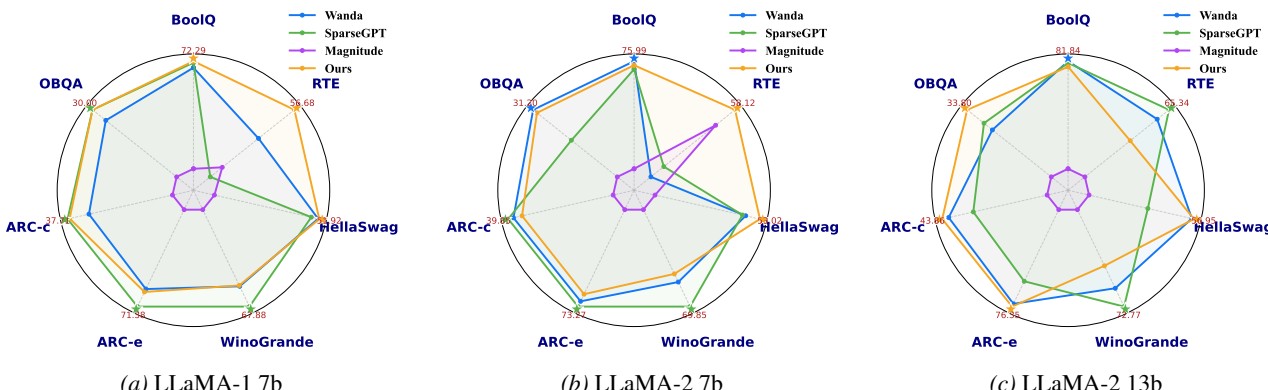

*(a)* LLaMA-1 7b     *(b)* LLaMA-2 7b     *(c)* LLaMA-2 13b

*Figure 2.* Our AutoPrune *vs.* peer competitors on 7 zero-shot tasks. (a) LLaMA-1 7b. (b) LLaMA-2 7b. (c) LLaMA-2 13b.

that our method consistently outperforms previous state-of-the-art competitors.

*Table 1.* Comparison of different reasoning strategies on **LLaMA 2 7B**. Metrics are reported on MMLU and Wikitext-V2 (%).

| Method | Step-by-step CoT | GCoT | MMLU | Wiki |
|---|---|---|---|---|
| Naive | ✗ | ✗ | 27.25 | 16.55 |
| Naive (step-by-step) | ✓ | ✗ | 29.15 | 7.17 |
| **AutoPrune (ours)** | ✓ | ✓ | **29.69** | **6.28** |

## 2. Rethinking the Design of LLMs Pruning

The main goal of unstructured pruning for LLMs is to learn a sparse and smaller subnetwork with minimal performance degradation through a training-free way. This raises a key problem: ***how can we accurately measure the importance of each weight?*** Naturally, we can leverage some statistical or theoretical properties of model weights or activations as an effective metric to obtain the importance of each weight. For instance, Wanda utilizes the magnitude of weights and input activations, enabling LLMs to effectively prune insignificant weights. However, those methods heavily rely on human experts (as shown in Table 2), which results in huge labor costs. Additionally, we validate that existing methods suffer from the outlier value issue, which leads to severe performance degradation under high pruning ratios. For the sake of clarity, we provide a series of strict validations to support our statements and reveal the root cause.

### 2.1. Experimentation Outlier Value Issue

To verify and understand the outlier value issue, we conduct a detailed experiment on LLaMA-1-7B by skewness sensitivity analysis. Unless otherwise specified, the AutoPrune search is performed only once on LLaMA-2-7B at 50% sparsity. The discovered pruning rule is then directly transferred to other model families, sparsity ratios, and downstream tasks without re-running the search for each setting. To be

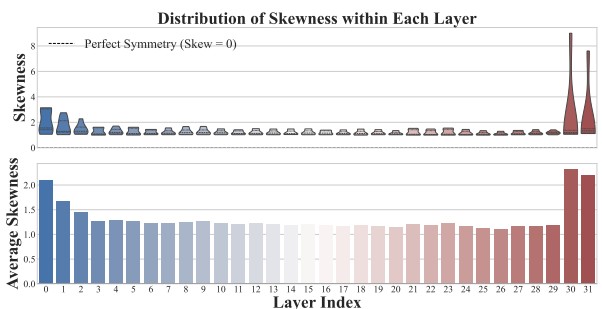

*Figure 3.* Validation on outlier value issue by skewness. Top: per-layer skewness. Bottom: mean skewness.

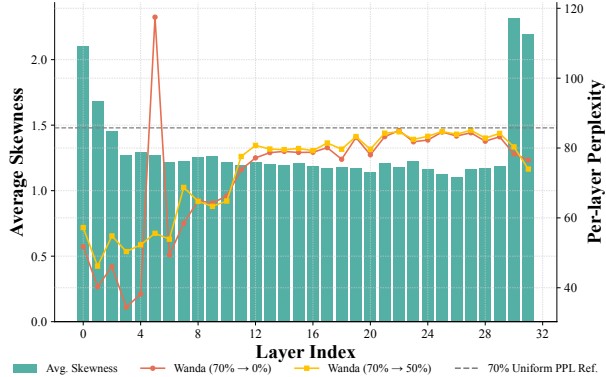

*Figure 4.* Validation of layer sensitivity to pruning ratios.

specific, we utilize skewness (see Eq. 4 for its definition) as a layer importance metric, accurately measuring skewness for the per-layer weights. As shown in Fig. 3, we show the distribution of skewness and average skewness for LLaMA-1-7B. Fig. 3 can yield the conclusion: Layers whose weight distributions exhibit stronger positive skewness tend to be more sensitive to pruning. Removing a small number of large weights can disproportionately degrade performance. For instance, removing the weights of "layer 1" at 70% will result in PPL increasing (from 45 to 95). In this paper, we define a layer with stronger positive skewness as "Outlier Value Issue".

*Table 2.* AutoPrune *vs.* previous methods. "$d$" denotes hidden dimension.

| Name | Formula | Human Expert | Calibration Data | Calibration Sample | Weight Update | Complexity |
|------|---------|:---:|:---:|:---:|:---:|:---:|
| Magnitude | $\|\mathbf{W}_{ij}\|$ | ✗ | ✗ | 0 | ✓ | $O(1)$ |
| SparseGPT | $[\|\mathbf{W}\|/diag[(XX^T + \lambda I)^{-1}]]_{ij}$ | ✓ | ✓ | 128 | ✓ | $O(d^3)$ |
| Wanda | $\|\mathbf{W}_{ij}\| \cdot \|X_j\|_2$ | ✓ | ✓ | 128 | ✗ | $O(d^2)$ |
| **AutoPrune (ours)** | $\frac{\|\mathbf{W}_{ij}\|}{\|\mathbf{W}_{i:}\|_1}\sqrt{\|X_{:j}\|_1 + \|X_{:j}\|_2^2}$ | ✗ | ✓ | 128 | ✗ | $O(d^2)$ |

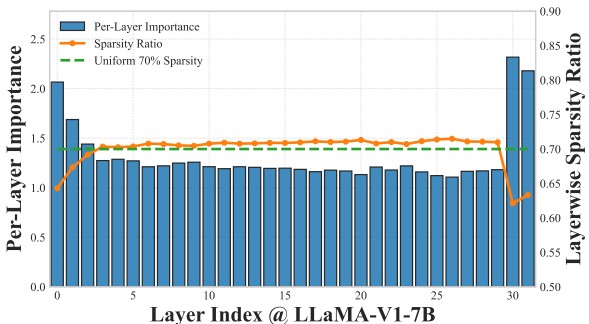

*Figure 5.* SDSA *vs.* Uniform allocation at 70% sparsity.

*Table 3.* Empirical validation of our SDSA with LLaMA-1/2 on WikiText at 60% high sparsity.

| Methods | LLaMA-1 7B | LLaMA-1 13B | LLaMA-2 7B | LLaMA-2 13B |
|---------|:---:|:---:|:---:|:---:|
| SparseGPT | 10.51 | 8.56 | 9.58 | 7.80 |
| SparseGPT + SDSA | **8.85** | **7.01** | **7.69** | **6.46** |
| Wanda | 10.66 | 8.56 | 9.71 | 7.75 |
| Wanda + SDSA | **9.88** | **7.65** | **8.89** | **6.98** |
| **AutoPrune** | 9.63 | 7.63 | 8.93 | 7.02 |
| **AutoPrune + SDSA** | **9.42** | **7.58** | **8.88** | **6.81** |

## 2.2. Further Analysis

**What factor of pruning causes the outlier value issue?**
To answer the above questions, we perform a series of strict validations using Wanda on LLaMA-1-7B in WikiText-2. To be specific, Fig. 4 shows the layer sensitivity to different pruning ratios, where experiments are conducted on LLaMA-1-7B in WikiText-2 via varying pruning ratios of layer $l_i$ to 0% and 50%, keeping others to 70%. As shown, the layer with the outlier value will result in high PPL (lower is better) at 70% high pruning ratios. By contrast, if we assign dynamic pruning ratios to those layers, the PPL will significantly decrease. In brief, outlier value is associated with uniform sparsity. To validate our statement, we conduct an additional experiment using Wanda on LLaMA-1-7B with our SDSA. As shown in Fig. 5, our SDSA provides a better dynamic sparsity. More validations are provided in **App. A.**

**Conclusion.** Consequently, the root cause of the outlier value issue can be attributed to uniform sparsity. This motivated us to design SDSA to mitigate the outlier value issue. As shown in Table 3, our SDSA successfully boosts the performance of all pruning methods (i.e., Wanda).

## 3. AutoPrune: Self-Pruning for LLMs

**Can LLMs Self-Design Pruning Algorithms?** We argue that LLMs are not only capable of this complex design task but are uniquely positioned to excel at it. LLMs possess vast, latent knowledge of network principles. They are trained on a colossal corpus of scientific literature, technical blogs, and open-source code, effectively internalizing decades of collective human knowledge on neural networks and optimization. The models' parameters hold a representation of what makes architectures work, what causes performance drops, and which patterns are redundant.

However, even advanced proprietary models such as GPT-o3 (OpenAI, 2025) and DeepSeek-R1 (Guo et al., 2025), which demonstrate strong emergent reasoning abilities, still suffer from significant limitations in reasoning. Even with reasoning capabilities, when these models face specific, specialized downstream tasks that demand structured and goal-directed thinking, they still inevitably run into problems with synthesizing information, making logical inferences, or handling multi-step planning. To address this gap, we introduce a Graph-driven Chain-of-Thought (GCoT) reasoning module. GCoT is explicitly designed to augment the reasoning ability of LLMs, providing a structured mechanism to guide pruning decisions through interpretable multi-step reasoning, thereby enabling our method to leverage the model's latent knowledge more effectively.

### 3.1. GCoT Driven Self-Pruning

**Chain of Thought.** To harness the LLM's potential, we design a Chain of Thought (CoT) pipeline that guides the model from a high-level goal to an implementable pruning algorithm. As illustrated in Fig. 6, the process includes four stages: **(1) Analysis**. the LLM analyzes model statistics to identify signals (e.g., weights, activations) relevant to pruning sensitivity; **(2) Hypothesis**. it synthesizes a concise causal statement linking these signals to parameter importance; **(3) Conceptual Formula**. the hypothesis is formalized into a semi-structured computational relation; and **(4) Computable Concept**. the relation is decomposed into modular, machine-readable components defining inputs, outputs, and operations.

This CoT procedure yields a complete, traceable algorithm, yet its linear nature restricts exploration to a single reasoning path. Suboptimal outcomes may arise simply because one trajectory fails to express a promising idea. This limitation motivates our Graph-driven CoT (GCoT), which systematically explores multiple reasoning branches to search for stronger pruning algorithms.

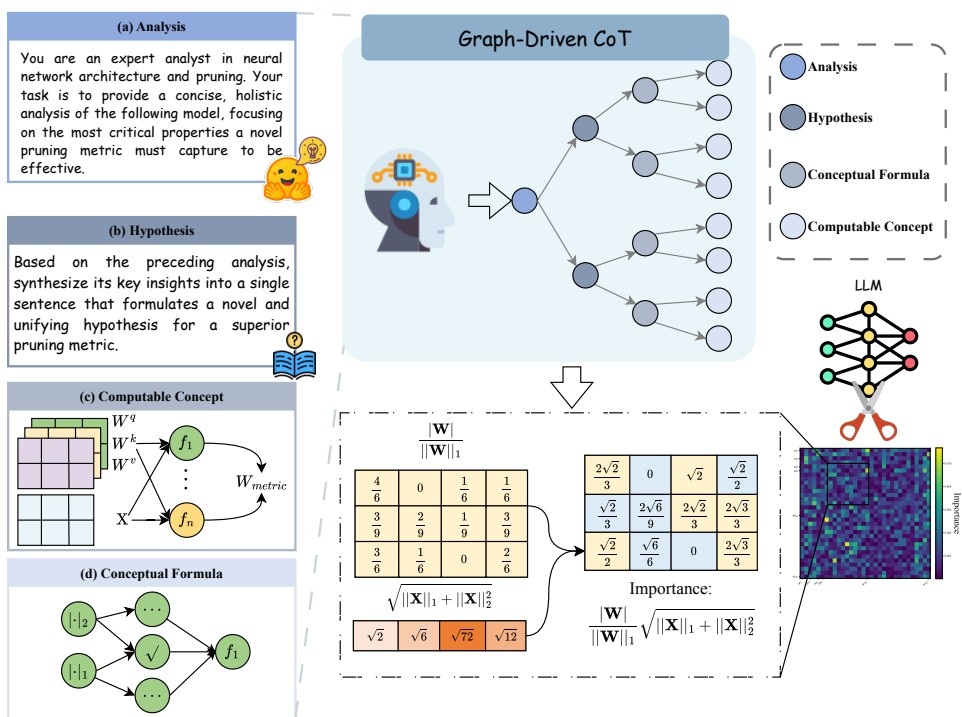

*Figure 6.* The intuitive framework of GCoT-driven self-pruning. This framework leverages an LLM to automatically generate pruning algorithms effectively, with the example below illustrating an optimal method discovered through this process.

**Graph-Driven CoT.** To explore a broader algorithmic space beyond a single reasoning trajectory, we generalize linear CoT into a directed acyclic reasoning graph. Let each node $s$ represent a reasoning state (e.g., *Analysis*, *Hypothesis*), and each edge be an LLM expansion step. At selected reasoning stages, we perform **temperature-based branching**:

$$\text{Expand}(s) = \{\text{LLM}(s; T_i)\}_{i=1}^k, \tag{1}$$

where $T_i$ is the sampling temperature and $k$ is the number of branches. It yields a graph $\mathcal{G}$ containing multiple distinct reasoning paths, and each root-to-leaf path $p$ corresponds to a complete pruning algorithm.

Each candidate algorithm induces an importance score function $I^{(p)}(W)$, which we evaluate by pruning the target model to sparsity $\alpha$ and computing perplexity on wikitext-2:

$$\mathcal{E} = \text{PPL}(\text{Prune}(W, I^{(p)}, \alpha)). \tag{2}$$

We then select the best algorithm by:

$$p^* = \arg\min_p \mathcal{E}(p). \tag{3}$$

This graph-driven process enables systematic exploration of diverse algorithmic structures while reusing shared reasoning components, improving both search efficiency and final pruning performance.

### 3.2. Skew-aware Dynamic Sparsity Allocation

Grounded in aforementioned observations of outlier value, we propose **Skew-Aware Dynamic Sparsity Allocation**

(SDSA), which combines (i) a simple yet effective skewness-indexed layer prior that adaptively reduces pruning in highly skewed outlier-bearing layers and (ii) a global sparsity-aware calibration term that tempers the skewness prior under low global sparsity, preventing both disproportionate pruning of some layers and excessive protection of others, and yielding a more stable adaptation trajectory.

**Skewness Measurement.** Skewness serves in our work as a quantitative indicator of the outlier value issue. Intuitively, layers whose weight magnitude distribution exhibits stronger positive skewness contain a small fraction of extremely large weights that disproportionately influence the model's output. For layer $\ell$, we compute the biased sample skewness of its weight distribution, intentionally omitting the small-sample correction since we analyze the LLMs weights rather than infer a population moment.

Let $W_\ell = \{w_{\ell,i}\}_{i=1}^{n_\ell}$ denote all weights in layer $\ell$. Because our pruning strategy depends on the prevalence of large-magnitude weights rather than the sign structure, we operate on the absolute values $A_\ell = \{|w_{\ell,i}|\}_{i=1}^{n_\ell}$. We quantify this asymmetry by the biased sample skewness:

$$\overline{w}_\ell = \frac{1}{n_\ell} \sum_{i=1}^{n_\ell} |w_{\ell,i}|, \quad m_{2,\ell} = \frac{1}{n_\ell} \sum_{i=1}^{n_\ell} (|w_{\ell,i}| - \overline{w}_\ell)^2,$$

$$\gamma_\ell = \frac{1}{n_\ell} \sum_{i=1}^{n_\ell} \left( \frac{|w_{\ell,i}| - \overline{w}_\ell}{\sqrt{m_{2,\ell}}} \right)^3. \tag{4}$$

Raw skewness magnitudes can vary across layers. To obtain a scale-stable index that is comparable across layers within a model, we center and range-normalize the $\{\gamma_\ell\}$ to $[-1, 1]$. Let:

$$\overline{\gamma} = \frac{1}{L} \sum_{\ell=1}^{L} \gamma_\ell, \quad \Delta\gamma = \max_j |\gamma_j - \overline{\gamma}|, \tag{5}$$

and define

$$\tilde{\gamma}_\ell = \frac{\gamma_\ell - \overline{\gamma}}{\Delta\gamma + \varepsilon}, \quad \tilde{\gamma}_\ell \in [-1, 1], \tag{6}$$

where $\varepsilon$ is a constant, typically set to 1e-8 to avoid division by zero. Positive $\tilde{\gamma}_\ell$ indicates a layer whose magnitude distribution is more positively skewed than the model-wide mean. Negative values indicate the opposite. We then convert the normalized skewness scores into relative protection weights through a softmax:

$$\omega_\ell = \frac{\exp(\beta\tilde{\gamma}_\ell)}{\sum_{j=1}^{L} \exp(\beta\tilde{\gamma}_j)}, \tag{7}$$

where $\beta$ is a coefficient that is dynamically modulated by the global sparsity level and skewness, and $\omega_\ell$ denotes the retention fraction for layer $\ell$.

**Global-Sparsity-Aware Schedule.** Based on the aforementioned insights, instead of hand-tuning $\beta$ directly, we devise an intuitive schedule to adapt $\beta$ automatically. We aim to ensure the ratio between the largest and smallest protection weights $\omega_{\max}$ and $\omega_{\min}$ remains bounded:

$$\frac{\omega_{\max}}{\omega_{\min}} = \exp\left(\beta\Delta\tilde{\gamma}\right) \leq M, \quad \Delta\tilde{\gamma} = \max_\ell \tilde{\gamma}_\ell - \min_\ell \tilde{\gamma}. \tag{8}$$

Solving for $\beta$ yields $\beta = \frac{\ln M}{\Delta\tilde{\gamma} + \varepsilon}$. To flatten allocations when global sparsity ratio $S_g \in [0, 1]$ is small, we modulate this contrast linearly: $\beta(S_g) = S_g \frac{\ln M}{\Delta\tilde{\gamma} + \varepsilon}$.

Thus, $\beta(0) = 0$ gives a uniform start, and $\beta(S_g)$ grows smoothly with the demanded pruning level, gradually releasing skewness sensitivity. In practice, we cap the allowable contrast with a modest value ($M$=1.8 for layerwise).

# 4. Experiments

We conduct the experimental validation of AutoPrune across multiple mainstream 7 LLMs, including LLaMA-1-(7B/13B/30B/65B) (Touvron et al., 2023a) and LLaMA-2-(7B/13B/70B) (Touvron et al., 2023b). For tasks and datasets, we evaluate AutoPrune's performance on the language modeling task in the WikiText dataset (Merity et al., 2016) and 7 zero-shot tasks from EleutherAI LM Harness (Gao et al., 2021). Notably, we use the perplexity on the

held-out WikiText and accuracy as evaluation metrics. We further include evaluations on the *From Passive to Active Reasoning* benchmark (Zhou et al., 2025) under a zero-shot setting, using GPT-4o (Hurst et al., 2024) as the response model and a 50%-pruned Llama-3-8B-Instruct (Meta AI, 2024) as the policy model to examine the generality of our reasoning framework beyond pruning scenarios. All experiments are conducted on an NVIDIA H100 GPU. The detailed experimental settings are presented in **App. B**.

## 4.1. Results on Language Modeling

Table 4 reports the perplexity of our method (AutoPrune without SDSA) and several strong baselines on pruned LLaMA-1 and LLaMA-2 models evaluated on the WikiText dataset. In this case, we observe that our method consistently outperforms existing training-free pruning approaches across various sparsity settings and model scales. To be specific, under 50% sparsity, our method achieves lower perplexity than the state-of-the-art training-free method Wanda across nearly all model sizes. For example, on LLaMA-1 13B and 30B, AutoPrune reduces perplexity from Wanda's 6.15 and 5.24 to 6.02 and 5.18, respectively, which represents absolute reductions of 0.13 and 0.06.

Under 60% sparsity, where the pruning task becomes more challenging, our method maintains its superiority. Compared to Wanda, AutoPrune reduces perplexity by 0.93 on LLaMA-1 13B (from 8.56 to 7.63), and by 0.75 on LLaMA-2 13B (from 7.75 to 7.02). These results indicate that our method is more robust to higher sparsity levels and better preserves model performance under aggressive compression. In the structured pruning settings (2:4 and 4:8 patterns), AutoPrune continues to outperform all baselines. In the 2:4 case on LLaMA-2 70B, AutoPrune reduces perplexity from Wanda's 5.16 to 4.97, an absolute improvement of 0.19. Similarly, in the 4:8 setting on LLaMA-1 30B, our method achieves 5.79 perplexity, outperforming Wanda's 5.97 and SparseGPT's 6.17.

## 4.2. Results on Zero-shot Tasks

To scrutinize the generalizability of AutoPrune (without SDSA), we conduct the experiments on different zero-shot downstream tasks with prompting, including BoolQ (Clark et al., 2019), RTE (Wang, 2018), HellaSwag (Zimmer et al., 2023), WinoGrande (Sakaguchi et al., 2019), ARC Easy and Challenge (Clark et al., 2018), and OpenBookQA (Mihaylov et al., 2018).

**LLaMA-1.** As shown in Table 5, we can clearly observe that our method achieves state-of-the-art performance across multiple zero-shot tasks under 50% unstructured sparsity. Notably, on the LLaMA-1 7B model, our method outperforms the SOTA training-free baseline "Wanda" by 1.08% on RTE and achieves the best accuracy of 72.29% on BoolQ.

*Table 4.* A performance comparison (lower is better) of pruned LLaMA-1 and LLaMA-2 models on the mainstream WikiText dataset. To make a fair comparison, we use "uniform" as the sample method used in previous methods (e.g., Wanda). Deep cyan, light cyan, and pink indicate the best, second-best, and good results, respectively.

| Methods | Weight Update | Sparsity | Human Expert | LLaMA-1 7B | LLaMA-1 13B | LLaMA-1 30B | LLaMA-1 65B | LLaMA-2 7B | LLaMA-2 13B | LLaMA-2 70B |
|---|---|---|---|---|---|---|---|---|---|---|
| Dense | - | 0% | | 5.68 | 5.09 | 4.77 | 3.56 | 5.12 | 4.57 | 3.12 |
| Magnitude | ✗ | 50% | ✓ | 17.29 | 20.21 | 7.54 | 5.90 | 14.89 | 6.37 | 4.98 |
| SparseGPT | ✓ | 50% | ✓ | 7.22 | 6.21 | 5.31 | 4.57 | 6.51 | 5.63 | 3.98 |
| Wanda | ✗ | 50% | ✓ | 7.26 | 6.15 | 5.24 | 4.57 | 6.42 | 5.56 | 3.98 |
| **AutoPrune** (ours) | ✗ | 50% | ✗ | 7.12↓(0.14) | 6.02↓(0.13) | 5.18↓(0.06) | 4.71 | 6.28↓(0.04) | 5.43↓(0.13) | 4.00 |
| Magnitude | ✗ | 60% | ✓ | 6e2 | 2e2 | 27.67 | 9.34 | 4e3 | 11.23 | 8.21 |
| SparseGPT | ✓ | 60% | ✓ | 10.51 | 8.56 | 6.66 | 5.82 | 9.58 | 7.80 | 4.98 |
| Wanda | ✗ | 60% | ✓ | 10.66 | 8.56 | 6.49 | 5.83 | 9.71 | 7.75 | 4.98 |
| **AutoPrune** (ours) | ✗ | 60% | ✗ | 9.63↓(0.88) | 7.63↓(0.93) | 6.31↓(0.18) | 5.79↓(0.03) | 8.93↓(0.65) | 7.02↓(0.75) | 4.78↓(0.20) |
| Magnitude | ✗ | 2:4 | ✓ | 42.13 | 18.37 | 9.10 | 7.11 | 54.59 | 8.33 | 6.33 |
| SparseGPT | ✓ | 2:4 | ✓ | 11.00 | 9.11 | 7.16 | 6.28 | 10.17 | 8.32 | 5.40 |
| Wanda | ✗ | 2:4 | ✓ | 11.53 | 9.58 | 6.90 | 6.25 | 11.02 | 8.27 | 5.16 |
| **AutoPrune** (ours) | ✗ | 2:4 | ✗ | 10.35↓(0.65) | 8.25↓(0.86) | 6.50↓(0.40) | 6.04↓(0.21) | 10.16↓(0.01) | 7.48↓(0.79) | 4.97↓(0.19) |
| Magnitude | ✗ | 4:8 | ✓ | 16.84 | 13.84 | 7.62 | 6.36 | 16.48 | 6.76 | 5.54 |
| SparseGPT | ✓ | 4:8 | ✓ | 8.61 | 7.40 | 6.17 | 5.38 | 8.12 | 6.60 | 4.59 |
| Wanda | ✗ | 4:8 | ✓ | 8.57 | 7.40 | 5.97 | 5.30 | 7.97 | 6.55 | 4.47 |
| **AutoPrune** (ours) | ✗ | 4:8 | ✗ | 8.14↓(0.43) | 6.89↓(0.51) | 5.79↓(0.18) | 5.29↓(0.01) | 7.50↓(0.47) | 6.16↓(0.39) | 4.45↓(0.02) |

*Table 5.* Accuracies (%) of LLaMA-1 for 7 zero-shot tasks with unstructured 50% sparsity. Because of the limited VRAM capacity of our available hardware, we restrict the 65B model to 8-bit quantization.

| Params | Methods | BoolQ | RTE | HellaSwag | WinoGrande | ARC-e | ARC-c | OBQA |
|---|---|---|---|---|---|---|---|---|
| | Dense | 75.05 | 66.43 | 56.92 | 69.93 | 75.34 | 41.89 | 34.40 |
| LLaMA-1 7B | Magnitude (Lee et al., 2021) | 54.59 | 54.51 | 45.49 | 59.19 | 58.84 | 33.53 | 22.40 |
| | SparseGPT (Frantar & Alistarh, 2023) | 72.05 | 54.15 | 51.43 | 67.88 | 71.38 | 37.71 | 30.00 |
| | Wanda (Sun et al., 2024) | 71.22 | 55.60 | 51.85 | 66.06 | 69.11 | 36.86 | 28.80 |
| | **AutoPrune** (ours) | 72.29 | 56.68 | 51.92 | 65.98 | 69.48 | 37.62 | 30.00 |
| | Dense | 77.89 | 70.4 | 59.94 | 72.77 | 77.40 | 46.50 | 33.20 |
| LLaMA-1 13B | Magnitude (Lee et al., 2021) | 54.89 | 51.26 | 44.16 | 63.14 | 58.80 | 33.79 | 27.20 |
| | SparseGPT (Frantar & Alistarh, 2023) | 76.97 | 61.01 | 54.95 | 71.67 | 72.47 | 41.98 | 31.20 |
| | Wanda (Sun et al., 2024) | 75.90 | 62.82 | 55.71 | 71.98 | 73.19 | 43.52 | 32.20 |
| | **AutoPrune** (ours) | 76.94 | 62.46 | 55.52 | 72.14 | 73.44 | 43.00 | 32.20 |
| | Dense | 82.69 | 66.79 | 63.35 | 75.69 | 80.30 | 52.82 | 36.00 |
| LLaMA-1 30B | Magnitude (Lee et al., 2021) | 64.34 | 50.18 | 50.59 | 66.54 | 72.39 | 43.77 | 29.00 |
| | SparseGPT (Frantar & Alistarh, 2023) | 82.32 | 62.45 | 59.15 | 75.22 | 78.96 | 48.56 | 35.00 |
| | Wanda (Sun et al., 2024) | 81.90 | 65.34 | 60.93 | 73.48 | 79.29 | 49.66 | 34.60 |
| | **AutoPrune** (ours) | 82.11 | 63.54 | 60.64 | 74.43 | 80.06 | 49.83 | 35.20 |
| | Dense | 84.83 | 69.68 | 64.54 | 77.27 | 81.40 | 52.90 | 38.20 |
| LLaMA-1 65B | Magnitude (Lee et al., 2021) | 79.15 | 62.45 | 61.90 | 74.74 | 76.40 | 49.57 | 35.00 |
| | SparseGPT (Frantar & Alistarh, 2023) | 84.60 | 70.76 | 63.90 | 77.43 | 79.35 | 50.85 | 37.20 |
| | Wanda (Sun et al., 2024) | 84.70 | 71.48 | 64.55 | 76.87 | 79.75 | 50.51 | 38.80 |
| | **AutoPrune** (ours, 8bit) | 84.79 | 72.56 | 61.76 | 75.77 | 79.89 | 50.97 | 38.80 |

*Table 6.* Accuracies (%) of LLaMA-2 for 7 zero-shot tasks with unstructured 50% sparsity.

| Params | Method | BoolQ | RTE | HellaSwag | WinoGrande | ARC-e | ARC-c | OBQA |
|---|---|---|---|---|---|---|---|---|
| | Dense | 77.74 | 62.82 | 57.17 | 68.90 | 76.39 | 43.52 | 31.40 |
| LLaMA-2 7B | Magnitude (Lee et al., 2021) | 63.00 | 57.04 | 49.13 | 63.30 | 64.10 | 34.64 | 26.80 |
| | SparseGPT (Frantar & Alistarh, 2023) | 75.02 | 54.15 | 52.37 | 69.85 | 73.27 | 39.85 | 29.20 |
| | Wanda (Sun et al., 2024) | 75.99 | 53.43 | 52.49 | 68.19 | 72.77 | 39.59 | 31.20 |
| | **AutoPrune** (ours) | 75.50 | 58.12 | 53.02 | 67.64 | 74.75 | 39.16 | 31.00 |
| | Dense | 80.52 | 65.34 | 60.06 | 72.22 | 79.42 | 48.46 | 35.20 |
| LLaMA-2 13B | Magnitude (Lee et al., 2021) | 57.61 | 55.96 | 54.40 | 65.27 | 70.54 | 38.40 | 27.80 |
| | SparseGPT (Frantar & Alistarh, 2023) | 81.44 | 65.34 | 55.83 | 72.77 | 74.83 | 42.24 | 32.60 |
| | Wanda (Sun et al., 2024) | 81.84 | 64.02 | 56.90 | 71.35 | 76.18 | 43.52 | 32.00 |
| | **AutoPrune** (ours) | 80.58 | 61.01 | 56.95 | 69.61 | 76.35 | 43.86 | 33.80 |
| | Dense | 83.40 | 67.87 | 66.10 | 78.06 | 82.55 | 54.44 | 37.20 |
| LLaMA-2 70B | Magnitude (Lee et al., 2021) | 70.55 | 60.65 | 61.50 | 73.48 | 75.70 | 49.23 | 35.40 |
| | SparseGPT (Frantar & Alistarh, 2023) | 83.55 | 70.40 | 63.80 | 78.85 | 82.40 | 53.75 | 38.20 |
| | Wanda (Sun et al., 2024) | 82.50 | 73.65 | 64.10 | 78.14 | 80.80 | 52.65 | 37.40 |
| | **AutoPrune** (ours, 8bit) | 83.60 | 71.48 | 64.65 | 78.89 | 81.20 | 52.90 | 37.60 |

*Table 7.* A comparison of different sparsity using Wanda.

| Sparsity/Perplexity | 30% | 40% | 50% | 60% | 70% | 80% |
|---|---|---|---|---|---|---|
| Global | 10293 | 10762 | 14848 | 17765 | 5147 | 39918.56 |
| ER-plus | 6.05 | 6.62 | 8.00 | 14.04 | 229.17 | 6013.91 |
| ER | 6.02 | 6.55 | 7.74 | 12.16 | 112.03 | 11151.18 |
| Uniform | 5.99 | 6.38 | 7.26 | 10.70 | 85.77 | 3499.88 |
| OWL | 6.01 | 6.42 | 7.41 | 12.04 | 75.06 | 1244.62 |
| **SDSA**(ours) | **5.98** | **6.33** | **7.13** | **9.88** | **66.93** | **892.35** |

*Table 8.* Ablation study for calibration samples.

| Models/samples | 16 | 32 | 128 | 512 | 1024 | 2048 |
|---|---|---|---|---|---|---|
| LLaMA-1 7b | 7.12 | 7.10 | 7.12 | 7.11 | 7.11 | 7.11 |

Moreover, on the larger 30B and 65B models, our method delivers highly competitive results, reaching 35.20% on OBQA (vs. 34.60% by Wanda) and 72.56% on RTE (vs. 71.48% by Wanda).

**LLaMA-2.** As shown in Table 6, the same conclusion can be drawn from evaluations on LLaMA-2 models: our method consistently delivers competitive or superior accuracy across various zero-shot tasks under 50% unstructured sparsity. For instance, on the LLaMA-2 7B model, our method achieves the best result on RTE with 58.12%, surpassing SparseGPT (54.15%) and Wanda (53.43%). Similarly, on the 13B model, we reach 56.95% on HellaSwag, outperforming all baselines. On the largest 70B model, our method attains the highest score on BoolQ (83.60%) and matches or closely trails top baselines on other tasks.

### 4.3. Effectiveness on Outlier Value Issue

As shown in Table 3, we can find that the performance of AutoPrune with SDSA surpasses that with Uniform at 60% sparsity. To further evaluate the effectiveness of our method, we compare SDSA with a broad range of baselines, including Uniform, ER, ER-plus, Global, and OWL (Yin et al., 2024a) on the WikiText dataset using LLaMA-1-7B under broad sparsity levels. As shown in Table 7, SDSA consistently achieves the lowest perplexity across all sparsity settings. Notably, we find that SDSA better mitigates the outlier value issue compared to its peer competitors (i.e., OWL). This is because SDSA is well-motivated and based on detailed skewness analysis for LLMs. Those results validate the effectiveness of our SDSA to outlier value issue, especially showing the superiority under high pruning ratios.

### 4.4. Ablation Study

**The impact of calibration samples :** We conduct an ablation study on LLaMA-1 7b in WikiText to study the impact of calibration samples using our method at 50% sparsity. As shown in Table 8, our method is non-sensitive to calibration samples, showing strong stability to calibration samples.

### 4.5. Results on Active Reasoning

We further evaluate our reasoning framework on the latest *"From Passive to Active Reasoning"* benchmark (Zhou et al., 2025) under a zero-shot setting. The response model is GPT-4o (Hurst et al., 2024), while the policy model is a Llama-3-8B-Instruct (Meta AI, 2024) pruned to **50% sparsity**. As shown in Fig. 7, structured reasoning methods (APD, SparseGPT) consistently outperform the naive baseline, demonstrating that our approach generalizes beyond model compression tasks.

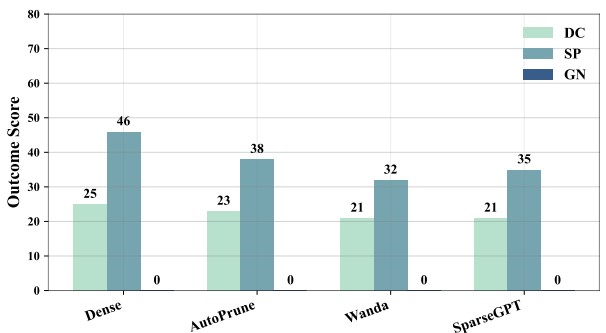

*Figure 7.* Results on the *From Passive to Active Reasoning* benchmark (Zhou et al., 2025) under zero-shot settings. DC and GN are evaluated by accuracy, while SP is measured by F1 score.

### 4.6. Additional Evaluation

Due to the page limit of the main text, we provide more results in **App. C-G**. ❶ Generalization for more LLMs and zero-shot tasks are shown in **App. C.** ❷ More ablation studies are shown in **App. D.** ❸ Detailed Prompt Engineering are shown in **App. E.** ❹ Limitations are shown in **App. F.** ❺ Related Work are shown in **App. G.**

## 5. Conclusion and Future Work

This paper identifies the outlier value issue that leads to severe performance drops in pruning large language models (LLMs) under high sparsity. Our method, AutoPrune, enables LLMs to automatically design optimal pruning algorithms without expert knowledge. By introducing Graph-driven Chain-of-Thought (GCoT) and Skew-aware Dynamic Sparsity Allocation (SDSA), AutoPrune effectively mitigates this issue and achieves superior pruning performance. We affirm that AutoPrune is practically valuable for efficient LLM pruning and hope it inspires further work on automated, adaptive sparsity design. In the future, we plan to extend Autoprune to more application scenarios, i.e., multimodal large models.

## Impact Statement

This paper presents work whose goal is to advance the field of Machine Learning. There are many potential societal consequences of our work, none which we feel must be specifically highlighted here.

## Acknowledgments

This work is supported by the Fundamental Research Funds for the Central Universities (N2623009).

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

# Overview of the Appendix

The overview of this appendix are as follows:

## A. More Validations on the Outlier Value Issue

To further validate our findings regarding the outlier value issue, which is attributed to uniform sparsity, we conducted a comprehensive sensitivity analysis. We used another three established pruning methods, SparseGPT, Magnitude, and our proposed AutoPrune, to evaluate the layer-wise sensitivity. Our results, which are detailed in Fig. 8, 9, and 10, clearly demonstrate a strong correlation between a layer's sensitivity to pruning and its weight distribution's skewness. Specifically, layers with higher skewness were found to be more susceptible to performance degradation when pruned, thus reinforcing our hypothesis that the outlier value issue is a critical factor to consider when designing effective LLM pruning strategies.

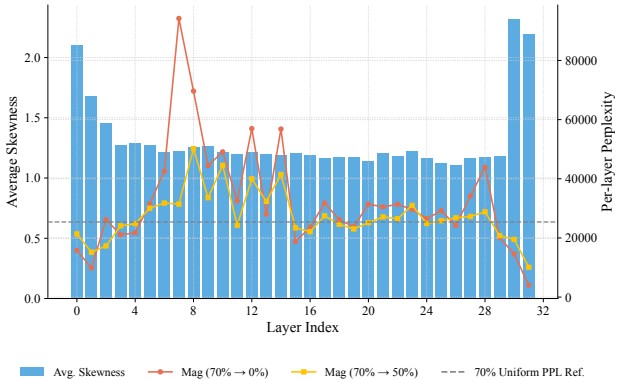

*Figure 8.* Validation of layer sensitivity to pruning ratios.

## B. Detailed Experimental Settings

**Search Protocol.** We provide the detailed search protocol of AutoPrune for reproducibility. Unless otherwise specified, the search is conducted once on LLaMA-2-7B under 50% unstructured sparsity, and the discovered pruning metric is then fixed and directly transferred to other model

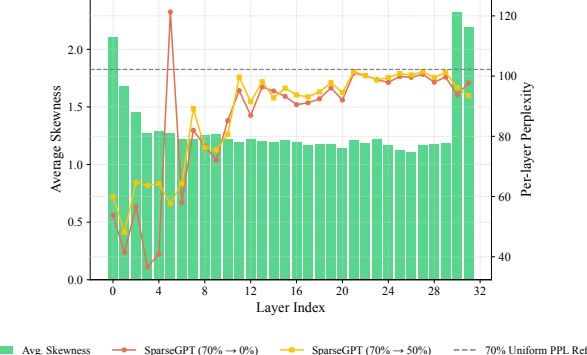

*Figure 9.* Validation of layer sensitivity to pruning ratios.

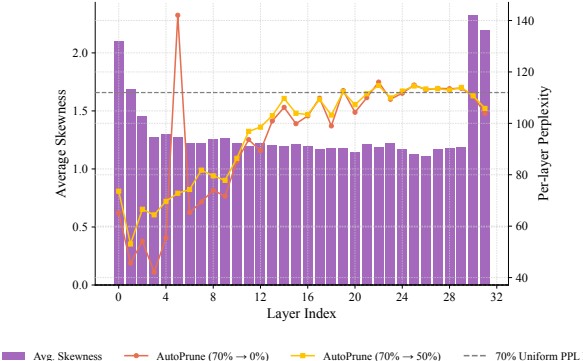

*Figure 10.* Validation of layer sensitivity to pruning ratios.

families, sparsity ratios, and downstream tasks without re-running the search. We use GPT-4o as the search LLM. In the Graph-driven Chain-of-Thought process, each reasoning stage expands $k = 5$ branches with temperatures $\{0.0, 0.3, 0.5, 0.7, 1.0\}$, producing approximately 625 candidate pruning metrics after the four reasoning stages. Each candidate is converted into an executable importance function, applied to prune the search model, and evaluated by the perplexity on the WikiText-2 training split. We rank all valid candidates by perplexity and select the candidate with the lowest perplexity as the final pruning metric. The selected metric is frozen for all reported experiments.

The full search requires about 780 LLM calls and approximately 6 GPU hours on a single NVIDIA H100 GPU. Candidate evaluation follows the same pruning protocol as our main experiments, using 128 calibration samples with sequence length 2048. Invalid candidates, including non-executable formulas or formulas with incompatible tensor shapes, are discarded before ranking.

**Note on the Comparison Scope.** Structured pruning methods (e.g., channel pruning or block removal) are not mainly included in our comparisons because they follow a fundamentally different acceleration mechanism. Such methods physically remove entire computation blocks, yielding

large speedups but typically causing considerable accuracy degradation, and are therefore not directly comparable to unstructured pruning approaches. In contrast, SparseGPT, Wanda, and our AutoPrune operate in the unstructured sparsity regime and rely on sparse matrix multiplication for acceleration. For a fair and meaningful evaluation, all baselines in this work are limited to unstructured pruning methods.

**Pruning Settings.** In this paper, our experimental settings are essentially identical to Wanda's (Sun et al., 2024), with the sole exception of the weight update method. We conducted all experiments on LLaMA-1 and LLaMA-2 models. For the pruning process, we used a seqlen of 2048 and nsamples of 128. Notably, while some prior methods like Wanda also implement a sequential weight update strategy, they often find the iterative approach to yield better results. In contrast, our method directly employs the more efficient sequential weight update and still achieves excellent results.

**SDSA.** For the Skew-Aware Dynamic Sparsity Dynamic Sparsity scheduler, the hyperparameter M is set to 1.8 for layer-wise allocation and 1.5 for block-wise allocation. We found that, contrary to the findings in OWL (Yin et al., 2024b), the layer-wise dynamic sparsity allocation in our experiments consistently yields superior results compared to the block-wise approach.

**Active Reasoning.** We follow the experimental protocol of From Passive to Active Reasoning (Zhou et al., 2025) and evaluate three task types: DC, GN, and SP. For DC and SP, we use a zero-shot active-reasoning setup based on a policy–response architecture, where llama-3-8B-Instruct (Meta AI, 2024) serves as the policy model to generate candidate reasoning turns and GPT-4o (Hurst et al., 2024) produces final responses. Both models use a temperature and top-p of 0.7, with a maximum of 25 turns.

## C. Additional Experimental Results

### C.1. Generalization for More LLMs

To further evaluate the generalizability of our method on large language models (LLMs), we conduct experiments on representative architectures: **Pythia-12B**. As shown in Table 9, AutoPrune consistently delivers strong performance across a wide range of sparsity levels, despite being applied to unseen model structures. Specifically, on Pythia-12B, AutoPrune matches or exceeds the performance of handcrafted baselines such as Wanda and SparseGPT across all sparsity levels, achieving a perplexity of 8.59 at 10% sparsity and maintaining low degradation even at 50% sparsity (11.13 vs. 11.27 for Wanda).

These results highlight the strong generalizability of AutoPrune to diverse LLMs, confirming its effectiveness as

a plug-and-play pruning algorithm that scales well across different transformer backbones without requiring weight updates or architecture-specific tuning.

## D. More Ablation Studies

### D.1. Number of Calibration Samples

In line with the experimental setup of Wanda (Sun et al., 2024), we use a calibration set of 128 samples. To further investigate the stability and robustness of our proposed method, we conduct a detailed analysis of its effect on the LLaMA and LLaMA-2 model families using this setting. We show the results for pruning LLaMA-7B and LLaMA-2-7B with unstructured 50% sparsity in Table 10. We find that there is a slight improvement in performance of pruned LLMs when the size of the calibration set goes beyond 128, but we maintain 128 for a fair and efficient comparison.

### D.2. Sensitivity Analysis on $M$

To evaluate the robustness of our method, we conduct a sensitivity analysis on the hyperparameter $M$. For this study, we apply 70% unstructured pruning on both the LLaMA-7B and LLaMA-2-7B models while varying the value of $M$. The results, summarized in Table. 11, demonstrates that our approach is not overly sensitive to the choice of M, as performance remains stable across the tested configurations.

### D.3. Number of Branches

To investigate the sensitivity of our AutoPrune method to its branches, we analyze the impact of the number of branches explored during the Graph-Driven Chain-of-Thought search process. We conduct experiments on the WikiText-2 dataset, where we apply 50% unstructured pruning to the LLaMA-1 7B model while varying the number of candidate branches. This study allows us to understand the trade-off between the breadth of the search for a pruning algorithm and the final performance of the pruned model. The perplexity results for different numbers of branches are presented in Table 12.

## E. Detailed Prompt Engineering

As detailed in the main text, our Graph-driven Chain-of-Thought framework utilizes four distinct prompts to guide the LLM through the progressive formalization of a new pruning algorithm. These prompts correspond to the four stages of the pipeline: Analysis, Hypothesis, Conceptual Formula, and Computational Concept. The specific prompts for each stage are as follows:

| Model | Dense | Pruning Method | Weight Update | Sparsity | | | | |
|---|---|---|---|---|---|---|---|---|
| | | | | 10% | 20% | 30% | 40% | 50% |
| Pythia-12B | 8.59 | Magnitude | × | 127.76 | 2e5 | 7e5 | 2e5 | 3e5 |
| | | SparseGPT | ✓ | **8.59** | 8.65 | 8.86 | 9.39 | **11.02** |
| | | Wanda | × | **8.59** | **8.60** | 8.85 | 9.31 | 11.27 |
| | | **AutoPrune** | × | **8.59** | **8.60** | **8.80** | **9.27** | 11.13 |

*Table 9.* Pruning Pythia-12B and OPT-13B with various sparsity levels.

| Model | Method | 1 | 16 | 32 | 64 | 128 | 256 | 512 | 1024 | 2048 |
|---|---|---|---|---|---|---|---|---|---|---|
| LLaMA-7B | SparseGPT | 10.22 | 7.61 | 7.36 | 7.29 | 7.26 | 7.20 | 7.19 | 7.23 | 7.20 |
| | Wanda | 7.46 | 7.27 | 7.28 | 7.28 | 7.26 | 7.30 | 7.26 | 7.25 | 7.26 |
| | **AutoPrune** | **7.15** | **7.12** | **7.11** | **7.12** | **7.11** | **7.11** | **7.11** | **7.11** | **7.11** |
| LLaMA-2-7B | SparseGPT | 8.63 | 6.67 | 6.62 | 6.61 | 6.53 | 6.52 | 6.50 | 6.49 | 6.49 |
| | Wanda | 6.53 | 6.45 | 6.46 | 6.45 | 6.45 | 6.45 | 6.45 | 6.45 | 6.45 |
| | **AutoPrune** | **6.33** | **6.30** | **6.29** | **6.29** | **6.28** | **6.29** | **6.29** | **6.28** | **6.28** |

*Table 10.* WikiText validation perplexity of pruned LLaMA-1 and LLaMA-2 under various number of calibration samples, with 50% sparsity.

---

**Stage 1: Analysis**

You are an expert analyst in neural network architecture and pruning. Your task is to provide a concise, holistic analysis of the following model, focusing on the most critical properties a novel pruning metric must capture to be effective.
TARGET MODEL FOR ANALYSIS:
{llm_description}
Based on the provided model's architecture and your expert knowledge, write a paragraph analyzing its core sensitivities and the most promising principles for effective pruning. Your analysis should identify the key trade-offs of simpler metrics and conclude by pointing towards the essential, multifaceted characteristics that a superior metric needs to measure. Provide only the analysis paragraph. Do not add any titles, introductory sentences, or concluding remarks.

**Stage 2: Hypothesis**

You are a principal scientist responsible for formulating novel research directions. Your task is to distill a detailed analysis into a single, powerful, and testable hypothesis.
PRECEDING ANALYSIS:
{analysis_text}
YOUR HYPOTHESIS TASK:
Based only on the preceding analysis, synthesize its key insights into a single sentence that formulates a novel and unifying hypothesis for a superior pruning metric.

The hypothesis should propose a clear relationship or dependency. For example: "A component's importance is a function of property X combined with property Y."
Provide only the single hypothesis sentence. Do not add any titles, introductory phrases, or explanations.

**Stage 3: Conceptual Formula**

You are a senior research engineer specializing in translating theoretical concepts into algorithmic structures. Your task is to convert a natural language hypothesis into a semi-formal conceptual formula.
PRECEDING HYPOTHESIS:
{hypothesis_text}
YOUR FORMALIZATION TASK:
Based on the provided hypothesis, create a conceptual formula that represents the core computational logic. This formula should be abstract, using mathematical notation and placeholders for functions and variables, to outline the structure without getting into implementation details.
For example, if the hypothesis is "Importance is the product of weight magnitude and activation norm," a valid conceptual formula would be: $Importance(W, X) = |W| * ||f(X)||$.
Provide only the conceptual formula. Do not add any titles, introductory sentences, or explanations.

| Model | 1.2 | 1.4 | 1.6 | 1.8 | 2.0 | 2.2 | 2.5 | uniform |
|---|---|---|---|---|---|---|---|---|
| LLaMA-7B | 99.30 | 90.45 | 81.06 | 79.94 | 79.01 | 82.26 | 87.24 | 111.98 |
| LLaMA-V2-7B | 99.09 | 97.26 | 95.42 | 95.19 | 95.46 | 93.60 | 100.03 | 127.92 |

*Table 11.* Sensitivity analysis of $M$, on the WikiText-2.

| Model | 1 | 3 | 5 | 10 | uniform |
|---|---|---|---|---|---|
| LLaMA-7B | 7.49 | 7.12 | 7.12 | 7.09 | 5.68 |
| LLaMA-V2-7B | 6.93 | 6.28 | 6.25 | 6.29 | 5.12 |

*Table 12.* Pruning LLaMA with various number of branches.

---

**Stage 4: Computational Concept**

You are a software architect designing a library of computational modules. Your task is to decompose a conceptual formula into a set of precise, machine-readable functional components.
CONCEPTUAL FORMULA:
{conceptual_formula}
YOUR DECOMPOSITION TASK:
Break down the conceptual formula into a series of modular, computable concepts. For each concept, define its precise functional signature, including a description, a list of inputs, and the expected output. This decomposition should create an unambiguous blueprint ready for direct implementation.
Provide only the list of computable concepts. Do not add any titles, introductory phrases, or concluding remarks.

---

## F. Limitations

While AutoPrune significawtly surpasses previous methods (i.e., SparseGPT, Wanda), some limitations remain. We organize them here for systematic discussion.

**Black-box Optimization** Although our method leverages an LLM to generate candidate pruning algorithms—rendering its internal token-level reasoning unobservable—we mitigate this black-box concern through a novel Graph-driven Chain-of-Thought (GCoT) framework. GCoT structures the prompt optimization process into a progressive reasoning pipeline, guiding the LLM from vague goals to precise, executable algorithm blueprints. Specifically, the pipeline comprises four explicit stages: (1) Analysis, where the LLM grounds its reasoning in model-specific statistics; (2) Hypothesis, where it proposes a causal relationship between observed signals and pruning sensitivity; (3) Conceptual Formula, which formalizes the logic in a structured form; and (4) Computable Concept, where this logic is modularized into machine-readable functions.

Crucially, all decisions that influence the search trajectory pass through this measurable, task-specific reasoning process, rather than being driven by opaque logits. By decomposing the algorithm search space into interpretable, modular components, GCoT exposes the full reasoning path behind each candidate algorithm. While the LLM itself remains a partially opaque system, the GCoT framework ensures that the optimization process is transparent, traceable, and grounded in explicit cognitive steps. This significantly enhances both the interpretability and trustworthiness of the learned pruning strategies.

**Security and Stability of Executing LLM-Generated Code** As LLM-generated proxies may contain unintended behaviors, including unsafe operations or excessive resource usage, executing them directly poses inherent risks. To mitigate these concerns, all code is executed within a controlled sandbox environment that enforces strict isolation, resource limits, and safety policies, ensuring stable and secure execution without compromising the host system.

## G. Related Work

**Layerwise Sparsity for Pruning.** Early approaches used uniform pruning (LeCun et al., 1989; Hu, 2016), where all layers are pruned at the same sparsity. However, (Frankle & Carbin, 2018) highlighted that different layers have varying importance, making this strategy suboptimal. To address this issue, some methods (Molchanov et al., 2019; 2016; Nonnenmacher et al., 2021; Zhang et al., 2022) automatically determine layerwise sparsity by selecting critical parameters across the entire network. Other studies (He et al., 2018; Yu et al., 2021) treat pruning as a search problem to identify layerwise sparsity. In contrast to these techniques for CNN models in vision tasks, our method focuses specifically on LLMs.

**LLM Pruning.** In contrast to traditional techniques, LLM pruning emphasizes data and time efficiency, meaning that pruned models do not require extensive retraining. LLM-Pruner (Ma et al., 2023) offers a structured pruning method based on model dependency relations and uses LoRA to recover the pruned model's performance. SparseGPT (Frantar & Alistarh, 2023) introduces an effective Hessian matrix estimation technique in large-scale models. In addition, Wanda (Sun et al., 2024) adopts a direct strategy based on the product of weights and activation values to eliminate weights. These methods apply a uniform pruning rate across

all layers. Recently, several approaches have been proposed to achieve non-uniform layerwise sparsity. BESA (Xu et al., 2024) employs a differentiable approach to search for optimal layerwise sparsity. DSA (Li et al., 2024a) utilizes a distribution function to allocate sparsity to layers, with the function being searched through evolutionary algorithms. ALS (Li et al., 2024b) develops an adaptive sparsity allocation strategy based on evaluating the relevance matrix using linear optimization. OWL (Yin et al., 2024b) linearly maps the non-outlier ratio of each layer to its sparsity. These methods rely on search processes or simple linear functions to derive sparsity or allocation strategies. However, for the high-dimensional search space of layerwise sparsity, they cannot guarantee achieving optimal solutions. To address this, we conduct dense empirical research to summarize LLM-specific pruning principles and propose a sparsity allocation strategy that satisfies these principles.

