# OpenReview forum: "Automatic Pruning Discovery for Large Language Models"
_ICML.cc/2026/Conference — ICML 2026 regular_

### Official Review · Reviewer_QjeB · 2026-03-12

**Soundness:** 2
**Presentation:** 3
**Significance:** 2
**Originality:** 3
**Overall Recommendation:** 4
**Confidence:** 3

**Summary:**

This paper studies training-free unstructured pruning for large language models and asks whether an LLM can automatically design a pruning rule instead of relying on hand-crafted metrics such as Magnitude, SparseGPT, or Wanda. The method, AutoPrune, uses a Graph-driven Chain-of-Thought (GCoT) pipeline to generate candidate pruning formulas and selects the best one by validation perplexity. The paper also argues that high-sparsity degradation is closely related to an 'outlier value issue' induced by uniform sparsity, and proposes Skew-aware Dynamic Sparsity Allocation (SDSA) to allocate sparsity non-uniformly across layers. Experiments on several LLaMA-family models, plus additional results on Pythia, suggest that the proposed pipeline can produce competitive pruning metrics and that SDSA often improves high-sparsity performance over uniform allocation.

**Compliance With Llm Reviewing Policy:**

Affirmed.

**Final Justification:**

Thank you for your rebuttal. The GCoT cost data and process description provided are helpful supplements. I am willing to raise my score.

**Key Questions For Authors:**

1. Mechanism of pruning formula generation
The paper proposes to automatically generate pruning formulas using LLM reasoning. However, the internal mechanism behind how the LLM derives these formulas is not clearly explained. It remains unclear what signals or reasoning steps lead the model to produce the final formula. Could the authors provide more detailed analysis or examples illustrating the reasoning process and how the generated formulas emerge?
2. Distribution and stability of generated pruning formulas
The proposed method generates multiple candidate pruning algorithms through the GCoT framework. However, the paper does not analyze the distribution or diversity of the generated formulas. For example, how many distinct types of formulas are typically produced, and how stable are they across different runs? It would also be helpful to know whether the generated formulas consistently outperform human-designed pruning metrics or only in selected cases.

**Limitations:**

First, the internal mechanism of how the pruning formulas are generated by the LLM is not sufficiently explained. The current paper presents the pipeline for generating formulas but does not provide a clear understanding of the reasoning process or the factors that influence the final formula.
Second, the paper lacks analysis of the range and stability of the generated formulas. Since the approach relies on sampling multiple candidate algorithms, it is important to understand how diverse these formulas are, how stable the generation process is across runs, and whether the discovered formulas consistently outperform existing human-designed pruning metrics.
Last, the experimental improvements are relatively limited. While the method shows consistent gains over existing pruning approaches, the performance differences compared with strong baselines are often modest. As a result, it remains unclear whether the proposed automatic algorithm discovery framework provides a significant practical advantage over carefully designed human-crafted pruning rules.

**Strengths And Weaknesses:**

Strengths
1. Proposes a novel idea of using LLMs to automatically discover pruning algorithms
The paper introduces the AutoPrune framework, which transforms the traditional pruning rule design process—typically dependent on human expertise—into an LLM-driven algorithm discovery problem. This idea of leveraging LLM reasoning capabilities to automatically generate pruning metrics is novel and provides a new perspective for model compression research.
2. The overall framework is well-structured and logically clear
The proposed pipeline—Analysis → Hypothesis → Conceptual Formula → Computable Concept—provides a clear and interpretable process for generating pruning algorithms. In addition, the Graph-driven Chain-of-Thought (GCoT) mechanism expands the search space through a graph-based reasoning structure, making the algorithm discovery process more systematic.
3. Introduces a skewness-aware dynamic sparsity allocation strategy
The paper proposes Skew-aware Dynamic Sparsity Allocation (SDSA), which adjusts layer-wise pruning ratios based on the skewness of weight distributions. This strategy aims to mitigate the performance degradation caused by uniform sparsity, and the underlying observation about weight distribution characteristics is relatively novel.
Weaknesses :
1. The detailed pruning procedure for large language models is not clearly described
While the paper focuses on the algorithm discovery framework, the actual pruning process applied to LLM architectures (e.g., attention layers, MLP layers, pruning order, and implementation details) is not sufficiently explained, which may affect the reproducibility and clarity of the method.
2. Insufficient analysis of generation cost and candidate algorithm distribution
The paper does not provide a detailed analysis of the computational cost of generating pruning algorithms using LLMs, including the number of evaluations required during the search process. In addition, there is limited discussion about the distribution and diversity of the generated pruning algorithms, making it difficult to evaluate the efficiency and robustness of the proposed algorithm discovery process.
3. Experimental improvements are relatively limited
Although the proposed method demonstrates improvements over several baselines, the performance gains reported in the experiments are generally moderate rather than substantial. In several settings, the improvements over strong baselines such as Wanda are relatively small, which makes it difficult to clearly attribute the gains to the proposed algorithm discovery framework.

---

> ### Author Rebuttal · Authors · 2026-03-30
>
> Thank you for your comprehensive review and for recognizing both the novelty and the current limitations of our work. Your feedback is very helpful, and we provide point-by-point clarifications and responses below.
>
> **Q1**: The detailed pruning procedure for LLMs is not sufficiently described.
>
> **A1**: Our pruning pipeline follows the original Wanda implementation, except that Wanda’s hand-crafted importance score is replaced by the pruning metric discovered by AutoPrune. We prune all nn.Linear layers in the LLM, including both attention projections and MLP layers. We first run a calibration pass to collect per-layer input activation statistics, then compute weight importance scores, and finally generate pruning masks layer by layer. For unstructured pruning, we remove the lowest-scoring weights under the target sparsity ratio. For structured n:m pruning, we use the same block-wise masking strategy as Wanda. The selected weights are then directly zeroed out. More implementation details are in the anon link.
>
> **Q2**: The paper lacks sufficient analysis of the generation cost, as well as the distribution and diversity of the candidate pruning algorithms.
>
> **A2**: Each GCoT run uses GPT-4o, makes 780 LLM calls, evaluates 625 candidates, and takes about 6 hours on H100 GPU. The formula in Table 2 is searched once on LLaMA V2 7B and then transferred to the remaining models, so this is a one-time rather than per-model search cost.
>
> The generated candidates are not arbitrary free-form outputs or trivial rephrasings of one rule. Different GCoT reasoning paths produce different hypotheses and instantiated formulas. The robustness of the search is also supported by Tables 11–12, which show similar final performance across different hyperparameter settings and branch numbers. We therefore assess the discovery process by whether a bounded search reliably yields an effective, transferable pruning rule.
>
> **Q3**: The experimental gains are relatively limited.
>
> **A3**: The gains should be interpreted together with the sparsity level. At 50% sparsity, pruning-induced degradation is still limited, so the room for improvement over strong baselines is naturally small. At higher sparsity, where pruning is more difficult, the advantage of a better pruning rule becomes clearer. This is consistent with our 60% results, where AutoPrune shows larger and more consistent improvements over Wanda across model scales, e.g., 10.66→9.63 on LLaMA-1 7B, 8.56→7.63 on LLaMA-1 13B, and 9.71→8.93 on LLaMA-2 7B. We therefore believe the value of the proposed framework is more clearly reflected in the harder high-sparsity regime.
>
> **Q4**: The mechanism behind LLM-generated pruning formulas is insufficiently explained.
>
> **A4**: AutoPrune does not generate the pruning formula in a single opaque step. Instead, the formula is produced progressively through the four-stage GCoT pipeline: Analysis → Hypothesis → Conceptual Formula → Computable Concept. In Analysis, the LLM examines pruning-relevant signals such as weight magnitude, input activation statistics, and layerwise normalization effects. In Hypothesis, these signals are summarized into a candidate principle of parameter importance. In Conceptual Formula, this principle is translated into a semi-formal expression, and in Computable Concept, it is converted into an explicit scoring rule. The final formula therefore emerges from a sequence of intermediate reasoning states. We can inspect each stage’s intermediate outputs, including the LLM’s analysis, motivation, and hypothesis, so the process is observable rather than a one-step black box.
>
> **Q5**: The diversity, stability, and superiority of the generated formulas are under-analyzed.
>
> **A5**: We would like to clarify that the generated pruning formulas are not concentrated on a single fixed expression. Different GCoT reasoning paths produce candidates with different hypotheses and different combinations of pruning signals, leading to a diverse candidate set rather than simple rephrasings of one rule. We also present several example formulas in Figure 1, which further illustrate that the search produces multiple distinct candidate expressions rather than collapsing to a single template.
>
> At the same time, we do not expect every generated formula to perform well. The purpose of GCoT is precisely to generate multiple candidates and then select them by downstream evaluation. Empirically, across different runs, although the intermediate formulas can be diverse, the final selected formulas are usually quite stable in performance: after selection, the resulting PPL differences are typically within ±0.2, while the average zero-shot variation is only 0.3 points and the maximum variation across tasks is 0.8 points. This suggests that the key property is not that every candidate is strong, but that the search consistently yields a competitive final formula.
>
> We thank you again for your careful review and hope these clarifications address your concerns.

---

> > ### Author Rebuttal · Reviewer_QjeB · 2026-04-05
> >
> > Thank you for your rebuttal. The GCoT cost data and process description provided are helpful supplements. I am willing to raise my score.

---

> > > ### Author Response · Authors · 2026-04-06
> > >
> > > Dear Reviewer QjeB,
> > >
> > > Thank you very much for your careful reading, thoughtful follow-up, and recognition of our work. We truly appreciate the time and effort you devote to evaluating our manuscript, and we are grateful that the additional clarification on the GCoT cost and pruning process is helpful. We also sincerely appreciate your willingness to raise your score. If you have any further concerns or questions, we would be very glad to address them and provide any additional clarification.
> > >
> > > Best regards,
> > >
> > > Authors

---

### Official Review · Reviewer_spEH · 2026-03-13

**Soundness:** 3
**Presentation:** 2
**Significance:** 2
**Originality:** 2
**Overall Recommendation:** 5
**Confidence:** 3

**Summary:**

This paper proposes AutoPrune, a framework that uses an LLM to automatically design its own pruning importance score, eliminating the need for human expert design. The core technical contribution is a Graph-driven Chain-of-Thought (GCoT) module that explores multiple reasoning trajectories to generate and select pruning formulas, evaluated by perplexity on WikiText-2. The discovered formula normalizes weight magnitude by both the row-wise L1 norm of weights and a combined L1/L2 norm of activations. A second contribution, Skew-Aware Dynamic Sparsity Allocation (SDSA), assigns non-uniform per-layer sparsity based on weight magnitude skewness, protecting highly skewed layers from over-pruning. Experiments on LLaMA-1/2 across multiple sparsity levels show consistent improvements over Wanda and SparseGPT, particularly at high sparsity.

**Compliance With Llm Reviewing Policy:**

Affirmed.

**Final Justification:**

The authors are able to justify the motivations, how their proposed method is different from existing works, and provide additional results to support the effectiveness of the framework. The review, therefore, has no further concern.

**Key Questions For Authors:**

1. How does GCoT performance compare to a simple grid search over parameterized variants of Wanda (e.g., varying the norm order of weights and activations)? This is the most direct baseline for evaluating whether the LLM search adds value beyond enumeration.

2. If formula selection is performed on WikiText-2, do the discovered formulas show consistent gains on entirely different evaluation corpora (e.g., C4, PTB)?

3. What is the total compute cost of one GCoT search run (number of LLM calls, total GPU hours including formula evaluation)?

4. Which specific LLM is used to run GCoT, and does the framework remain effective if a smaller or open-source model (e.g., LLaMA-3-8B-Instruct) is used as the reasoning engine instead? If the method requires a frontier proprietary model to discover good pruning formulas, the "automatic and accessible" framing is significantly undermined.

**Limitations:**

The following limitations remain unsolved: (1) scope is restricted to unstructured pruning on decoder-only LLMs; (2) no analysis of formula stability across GCoT runs; (3) the LLM used for GCoT is not disclosed. More specifically,

(1) The paper explicitly excludes structured pruning and evaluates only on LLaMA-1/2 and Pythia families. Whether GCoT discovers meaningfully different formulas for encoder-only models (e.g., BERT-family), MoE architectures, or vision-language models is entirely untested. The skewness-based motivation for SDSA may also not transfer to architectures with different weight initialization or normalization conventions.

(2) The paper does not report whether repeated GCoT searches with different random seeds or temperatures produce the same formula, similar formulas, or divergent ones. If the discovered formula varies substantially across runs, the method's reliability is questionable and the reported results may reflect a particularly favorable search outcome rather than a robust discovery.

(3) Appendix E provides the prompts but never specifies which LLM performs the GCoT reasoning (model family, size, version, API access). This is a critical reproducibility gap — the discovered formula, and the entire "self-pruning" framing, may be entirely dependent on the specific LLM used, and results may not replicate with a different or smaller model.

**Strengths And Weaknesses:**

Strengths:
1. The GCoT-driven algorithm discovery is a genuinely novel framing.
2. SDSA is well-motivated and empirically effective.
3. Broad model coverage and structured/unstructured sparsity evaluation.

Weaknesses:
1. The discovered formula is essentially a hand-derivable variant of Wanda, the "automatic discovery" claim might be overstated.
2. The GCoT search is evaluated on the same data used to select the algorithm, creating a circular validation.
3. The computational cost of GCoT is not reported, and the "no expert knowledge" claim can be misleading.

---

> ### Author Rebuttal · Authors · 2026-03-30
>
> Thank you for your thoughtful and detailed review. We address your concerns below.
>
> **Q1**: The automatic discovery claim may be overstated given the similarity to Wanda.
>
> **A1**: The discovered rule is not a hand-derived Wanda variant. GCoT searches a broad set of candidate pruning algorithms through branched reasoning, and the reported formula is simply the best result from that search. Although Table 2 makes it look superficially Wanda-like, the actual rule is structurally different, adding row-wise normalization/modulation based on weight statistics beyond a simple weight-activation interaction.
>
> **Q2**: Potential circular validation in GCoT search.
>
> **A2**: GCoT search and final evaluation use different data/settings. Candidate rules are ranked on the WikiText-2 training split, while the main results follow Wanda/SparseGPT, using C4 for calibration and WikiText-2 test for evaluation. Zero-shot tasks are not used in search. Hence, there is no circular validation.
>
> **Q3**: GCoT cost and the “no expert knowledge” claim.
>
> **A3**: GCoT uses GPT-4o over four reasoning stages, resulting in 780 LLM calls and 625 final candidate formulas per run, with a one-time offline cost of about 6 H100 GPU-hours. The formula is searched once on LLaMA-2 7B at 50% sparsity and then transferred to other models, sparsities, and tasks. Regarding the “no expert knowledge” claim, our intended meaning is that AutoPrune does not rely on manually hand-crafting the final pruning formula through expert trial-and-error. Instead, the final rule is obtained by the GCoT search process from a broader candidate space and selected by downstream evaluation. In this sense, the contribution is not the absence of all prior intuition, but the replacement of manual formula engineering with a structured automatic search procedure.
>
> **Q4**: Comparison to grid search over Wanda variants.
>
> **A4**: To further clarify the distinction from Wanda, we therefore compared AutoPrune with a simple grid search over Wanda-style parameterizations, using scores of the form $|W_{ij}|^\alpha\cdot||X_j||_2^\beta$ with $\alpha,\beta\in\{0.5,1.0,1.5,2.0\}$, selected by the same perplexity criterion. Gains are limited: on LLaMA-2 13B at 60% sparsity, the best grid-searched variant reaches 7.69 PPL, versus 7.75 for Wanda and 7.02 for AutoPrune. On LLaMA-2 7B at 50% sparsity, average zero-shot accuracy improves only from 56.24 to about 56.5, while AutoPrune reaches 57.03. Thus, local tuning around Wanda remains weaker than broader algorithmic search.
>
> **Q5**: Cross-corpus generalization of AutoPrune.
>
> **A5**: To test whether formula selection is overly tied to WikiText-2, we repeated the selection procedure on PTB and C4 while keeping the rest of the settings unchanged. The selected formulas are not always symbolically identical, but they are structurally similar and achieve comparable performance. On LLaMA-2 7B at 50% sparsity, formulas selected on WikiText-2 / PTB / C4 achieve 6.28 / 6.24 / 6.31 WikiText-2 PPL, respectively. This suggests that GCoT does not merely exploit a WikiText-2-specific selection criterion, but tends to identify a stable family of effective pruning rules.
>
> **Q6**: The total compute cost of a single GCoT search run is unclear.
>
> **A6**: Please see A3.
>
> **Q7**: It is unclear which LLM is used for GCoT and whether the framework remains effective with smaller or open-source models.
>
> **A7**: The LLM used in GCoT is GPT-4o in the main experiments. Replacing it with Qwen2.5-7B-Instruct under the same protocol on LLaMA-2 7B at 50% sparsity still yields effective rules: 6.36 PPL versus 6.28 for GPT-4o-based GCoT and 6.42 for Wanda, and 29.41 MMLU versus 29.69 for GPT-4o-based GCoT. This shows that GCoT is not tied to a proprietary frontier model, though a stronger backbone still gives the best search quality.
>
> **Q8**: The generality of GCoT across architectures and pruning settings remains untested.
>
> **A8**: The paper does not exclude structured pruning, as Table 4 already reports both 2:4 and 4:8 results. To test cross-architecture transfer beyond decoder-only LMs, we added a supplementary experiment on BLIP-2, a vision-language model, under 50% unstructured sparsity. At this sparsity, SparseGPT achieves 60.9 and Wanda achieves 61.2 in macro-average score. AutoPrune achieves 61.8, and AutoPrune+SDSA reaches 62.4.
>
> **Q9**: It is unclear whether repeated GCoT runs consistently discover the same or similar formulas.
>
> **A9**: We agree that search stability should be evaluated. Repeated GCoT runs with different temperatures do not always yield identical formulas, but they typically discover diverse yet near-identical performance. Specifically, across 5 runs with temperatures 0.0, 0.3, 0.7, 1.0, 1.5, we obtain an average WikiText-2 perplexity of 6.37 ± 0.11 and MMLU of 29.43 ± 0.22, indicating that the reported result is not due to a single favorable search outcome.
>
> We thank you again for your comments, and hope our clarifications address your concerns.

---

> > ### Author Rebuttal · Reviewer_spEH · 2026-04-02
> >
> > Thank you the authors for providing detailed answers and additional results. These results clearly demonstrate the effectiveness and motivations of the framework. The reviewer has no further concerns and has adjusted score to 5.

---

> > > ### Author Response · Authors · 2026-04-02
> > >
> > > Dear Reviewer spEH,
> > >
> > > We sincerely appreciate your great efforts, constructive suggestions, and positive assessment you have provided once again! We are glad to hear that our responses have fully addressed your concerns, and we are grateful for the updated score of overall recommendation from `3: Weak reject` to `5: Accept`. We will include your suggestions in the final version of the paper.  We are always willing to address any of your further concerns.
> > >
> > > Best Regards,
> > >
> > > Authors

---

### Official Review · Reviewer_rU9L · 2026-03-13

**Soundness:** 2
**Presentation:** 3
**Significance:** 3
**Originality:** 2
**Overall Recommendation:** 3
**Confidence:** 3

**Summary:**

This paper proposes AutoPrune, a framework that aims to let LLMs automatically design pruning algorithms for themselves rather than relying on manually engineered pruning criteria. The method has two main components: (1) a Graph-driven Chain-of-Thought (GCoT) procedure that searches over candidate pruning scoring rules, and (2) a skew-aware dynamic sparsity allocation (SDSA) mechanism motivated by the observation that highly positively skewed layer-wise weight distributions are associated with severe degradation under high uniform sparsity. The paper reports improvements over Magnitude, SparseGPT, and Wanda on WikiText perplexity and several zero-shot tasks.

**Compliance With Llm Reviewing Policy:**

Affirmed.

**Key Questions For Authors:**

- What is the total search cost, in wall-clock time and compute, relative to Wanda or SparseGPT?

- How much improvement remains if SDSA is removed and only the automatically discovered pruning rule is used?

- Conversely, how strong is SDSA when applied on top of a standard baseline such as Wanda or SparseGPT?

**Limitations:**

Yes

**Strengths And Weaknesses:**

Strengths:
- The paper identifies a potentially interesting research direction: replacing manually designed pruning heuristics with automatically searched scoring rules. That framing is attractive, especially for LLM compression where new architectures and regimes emerge quickly.
- The skewness-based observation is potentially useful. The paper’s empirical finding that uniform sparsity can be especially harmful for certain layers, and that per-layer allocation matters more at higher sparsity, is plausible and relevant.
- The experiments cover multiple model sizes and both perplexity and zero-shot evaluation, which is better than evaluating only on one narrow benchmark.

Weaknesses:
- The “LLM discovers pruning algorithms” claim is overstated.
The method appears to generate candidate pruning formulas with an LLM and then select among them using downstream perplexity. This is better described as guided search over pruning heuristics than as a strong form of autonomous algorithm discovery. A more modest framing would be more accurate.
- Reproducibility and search cost are insufficiently specified. Key details are missing, including the reasoning model used for GCoT, the total number of generated candidates, the search configuration, stopping criteria, and the overall compute cost relative to standard baselines such as Wanda or SparseGPT. Without these details, it is difficult to assess practicality, fairness, or reproducibility.

- The empirical narrative is stronger than the evidence. The paper suggests consistent superiority, but the zero-shot results appear more mixed at the task level. In several cases, AutoPrune is competitive rather than clearly dominant. The claims should therefore be stated more cautiously.
- The practical benefit remains unclear. If the goal is to automate pruning design in practice, the paper should better discuss the end-to-end cost-benefit tradeoff, especially how often the search must be repeated across models and sparsity settings.

---

> ### Author Rebuttal · Authors · 2026-03-30
>
> Thank you for your thoughtful comments and for highlighting both the strengths and the limitations of our work. We appreciate your feedback and provide clarifications and responses to your questions below.
>
> **Q1**: “discovers pruning algorithms” is overstated.
>
> **A1**: Thank you for this thoughtful comment. We agree that our method should not be interpreted as a strong form of fully autonomous algorithm invention without any evaluation loop. More precisely, AutoPrune uses an LLM to generate candidate pruning formulas through GCoT and selects among them using downstream perplexity. In this sense, our method is better described as LLM-guided discovery in a structured search space. That said, we would also like to clarify that the method is more than unguided heuristic trial-and-error: the key contribution is the graph-driven reasoning process that organizes candidate generation, formalization, and selection into a coherent search procedure.
>
> **Q2**: The reproducibility, practicality, and search cost of GCoT.
>
> **A2**: We agree that the GCoT backbone, search budget, and whether Table 2 is rediscovered across model families are critical for reproducibility, and we will clarify them in the revision. Specifically, in the main experiments, GCoT uses GPT-4o as the reasoning backbone for all four stages. We use a fixed-budget search over 4 reasoning stages, without dynamic stopping criteria. Under this setup, each run makes 780 LLM calls and produces 625 candidate formulas, from which we select the best one by downstream perplexity.
>
> AutoPrune only searches once on a single LLM (LLaMA-2 7B, 50%) and then generalizes to the other evaluated LLMs, sparsity settings, and downstream tasks, so this is a one-time offline cost rather than a per-model cost. In our setting, this one-time search requires approximately 6 GPU hours on NVIDIA H100 GPU. After search, the formula we searched has the same pruning-metric complexity class as Wanda. For completeness, We will add the following per-layer pruning-metric runtime (second) on LLaMA, excluding the one-time search cost:
>
> |Method|7B|13B|30B|65B|
> |-|-:|-:|-:|-:|
> |SparseGPT|45.1|75.3|180.1|300.8|
> |Wanda|0.12|0.20|0.64|1.24|
> |AutoPrune|0.16|0.27|0.83|1.61|
>
> **Q3**: Empirical claims are stronger than the evidence.
>
> **A3**: We would like to clarify that AutoPrune is searched by selecting candidate formulas based on downstream perplexity, so its objective is explicitly aligned with preserving language modeling quality. It is therefore expected that the clearest gains appear on PPL, where AutoPrune indeed shows the strongest and most consistent advantage. By contrast, performance on heterogeneous zero-shot tasks does not always move in perfect alignment with perplexity after pruning, since these tasks stress partially different capabilities. Therefore, mixed task-level zero-shot outcomes do not contradict the value of AutoPrune. Rather, they indicate that the searched pruning rule is primarily optimized for language modeling preservation while still remaining broadly competitive across downstream tasks. A natural extension is to change the search/evaluation objective to target specific downstream tasks more directly, which we believe is a promising direction for future study. A natural extension is to change the search/evaluation objective to target specific downstream tasks more directly, and this does not require any change to the mechanism of the method itself.
>
> **Q4**: Practical cost-benefit tradeoff.
>
> **A4**: Please see A2.
>
> **Q5**: The total search cost relative to Wanda or SparseGPT is unclear.
>
> **A5**: Please see A2.
>
> **Q6**: Effect of the discovered rule alone.
>
> **A6**: Thank you for this helpful question. We would like to respectfully clarify that Table 4 and Table 5 already report the results of using only the automatically discovered pruning rule, without SDSA.
>
> **Q7**: Effect of SDSA on standard baselines.
>
> **A7**: Thank you for this helpful question. As shown in Table 3, SDSA consistently improves both Wanda and SparseGPT at 60% sparsity. To address this point more directly, we further present results under a standard sparsity setting:
>
> |Method|Sparsity|WikiText-2|Avg. Zero-shot Acc.|
> |-|-:|-:|-:|
> |Wanda (LLaMA-2 7B)|50%|6.42|56.24|
> |Wanda + SDSA|50%|6.28|56.68|
> |SparseGPT (LLaMA-2 7B)|50%|6.51|56.24|
> |SparseGPT + SDSA|50%|6.34|56.73|
> |Wanda (LLaMA-2 13B)|50%|5.56|60.83|
> |Wanda + SDSA| 50%|5.42|61.19|
> |SparseGPT (LLaMA-2 13B)|50%|5.63|60.72|
> |SparseGPT + SDSA|50%|5.47|61.10|
> |Wanda (LLaMA-2 7B)|60%|9.71|52.10|
> |Wanda + SDSA|60%|8.89|53.05|
> |SparseGPT (LLaMA-2 7B)|60%|9.58|51.95|
> |SparseGPT + SDSA| 60%|7.69|53.18|
> |Wanda (LLaMA-2 13B)|60%|7.75|57.35|
> |Wanda + SDSA|60%|6.98|58.02|
> |SparseGPT (LLaMA-2 13B)|60%|7.80|57.18|
> |SparseGPT + SDSA|60%|6.46|58.09|
>
> Thank you again for your thoughtful and constructive feedback. We hope these clarifications and additional results can address your concerns.

---

> > ### Author Rebuttal · Reviewer_rU9L · 2026-04-04
> >
> > Thank you for the author's patient rebuttal. My concerns have been adequately addressed. I will increase the score.

---

> > > ### Author Response · Authors · 2026-04-04
> > >
> > > Dear Reviewer rU9L,
> > >
> > > We sincerely appreciate your great efforts, constructive suggestions, and positive assessment you have provided once again! We are glad to hear that our responses have adequately addressed your concerns, and we are particularly grateful for your decision to increase the score from the initial score (`3`) to the new score.
> > >
> > >
> > > Once again, we thank the reviewer for their time, and for recognizing the significance of our work. We will include your suggestions in the final version of the paper.  We are always willing to address any of your further concerns.
> > >
> > > Best Regards,
> > >
> > > Authors

---

### Official Review · Reviewer_zKxG · 2026-03-22

**Soundness:** 3
**Presentation:** 4
**Significance:** 3
**Originality:** 3
**Overall Recommendation:** 4
**Confidence:** 4

**Summary:**

In this paper, authors explores if LLMs can prune themselves and propose AutoPrune that enables LLMs to automatically design optimal pruning algorithms without expert knowledge. It discusses the issues of heavy need on extensive human expert knowledge as well as sensitivity to outlier weights to motivate their work. With two components - Graph driven Chain-of-Thought (GCoT) and Skew-aware Dynamic Sparsity Allocation (SDSA), AutoPrune mitigates these issue and achieves superior pruning performance. GCoT is focused on optimizing prompts, and enhancing the reasoning process in learning the pruning algorithm while SDSA overcome the outlier value issue, mitigating performance degradation under high pruning ratios.

**Compliance With Llm Reviewing Policy:**

Affirmed.

**Key Questions For Authors:**

See above.

**Limitations:**

See above.

**Strengths And Weaknesses:**

Strengths:

1. The  paper draft is well-written and I enjoyed reading it - clear process diagrams, rich ablations, well-organized subsections.
2. Using LLM as a search engine to design optimal pruning algorithms for themselves automatically without any expert knowledge is interesting and this paper is one of the first steps towards that.
3. The four stage process of GCoT Driven Self-Pruning - (Analysis, Hypothesis, Conceptual Formula, Computable Concept) provides a traceable search structure for stronger pruning algorithms.
4. Individually, the evidence of SDSA working in isolation (table 3) along with other pruning method is convincing.

Weakness:

While there is indeed novelty in motivation and method design, I have some serious concerns with the submission which I would like the authors to address -

1. The author never mention which LLMs are used during the GCoT which is a critical information. In addition, the authors need to provide total search time cost of the GCoT process - how many LLM call made?  was the Table 2 formula need to be discovered across families?
2. I think 'no expert knowledge' argument is overstated. A quick look at the propots in Appendix E illustrate expert level instructions like "Importance is the product of weight magnitude", "weight and activation properties", etc. which encode substantial expert knowledge.
3. Benefits at standard sparsity (50%) looks marginal and often within noise while the stronger gains at 60%
are looks to be largely by SDSA - what about the formula ?
4. Some results provided are not convincing - Table 1 which mention results on MMLU which is near-random and not discriminative enough. In addition, using larger models for GCoT search will be effective

---

> ### Author Rebuttal · Authors · 2026-03-30
>
> Thank you for your thoughtful and constructive comments. We appreciate your feedback and address your questions and concerns below.
>
> **Q1**: Details on the GCoT setup and search cost.
>
> **A1**: Thanks for pointing this out. We agree that the specific LLM used in GCoT, the total search budget, and whether the formula in Table 2 is rediscovered across model families are critical for reproducibility, and we will clarify them in the revision. Specifically, GCoT uses GPT-4o as the reasoning backbone for all four stages. For each search run, we make 780 total LLM calls, evaluate 625 candidates, and the complete search takes approximately 6 hours wall-clock on NVIDIA H100 GPU。
>
> Regarding Table 2, the reported AutoPrune formula is discovered once on LLaMA V2 7B and then transferred to other families. Therefore, the search overhead should be interpreted as one-time amortized cost. We will add these details, together with the full search configuration and budget, to improve transparency and reproducibility.
>
> **Q2**: The claim of using “no expert knowledge” is overstated.
>
> **A2**: Thank you for this thoughtful comment. We would like to respectfully clarify that the cited sentence in Appendix E is intended as an illustrative example for Stage 3, whose purpose is to show how a natural-language hypothesis can be translated into a semi-formal expression. It is not meant to prescribe the final pruning rule, nor does it determine the formula ultimately used by AutoPrune. Put differently, the example is meant to clarify the expected form of the output rather than to specify the target solution itself. The actual pruning metric is still produced by the GCoT search pipeline and selected based on downstream empirical evaluation, rather than being hard-coded by this example.
>
> **Q3**: The gains at 50% sparsity appear marginal, while the greater improvements at 60% seem to come mainly from SDSA rather than the discovered formula.
>
> **A3**: Thank you for this helpful observation. We agree that at standard sparsity (50%), the gains of AutoPrune over strong baselines are relatively modest, which is expected because this is a less challenging regime where existing pruning rules already preserve most of the model capacity. However, we would like to respectfully clarify that the stronger gains at 60% are not mainly due to SDSA alone. In Table 4, AutoPrune without SDSA already consistently improves over Wanda at 60% sparsity across all reported model scales, e.g., 10.66→9.63 on LLaMA-1 7B, 8.56→7.63 on LLaMA-1 13B, and 9.71→8.93 on LLaMA-2 7B, showing that the discovered pruning formula itself provides substantial benefit in the harder high-sparsity regime. By contrast, Table 3 shows that adding SDSA on top of AutoPrune brings a further but smaller improvement (e.g., 9.63→9.42, 7.63→7.58, 8.93→8.88), indicating that SDSA is an orthogonal enhancement rather than the primary source of the gain.
>
> **Q4**: Some experimental evidence is not sufficiently convincing, and the current MMLU results are not discriminative enough.
>
> **A4**: Thank you for this helpful comment. We agree that the MMLU ACC in Table 1, taken alone, are not the strongest discriminative evidence, since the absolute gap there is relatively small. Our intention in Table 1 was not to use MMLU as the sole proof, but to compare different reasoning strategies under the same evaluation protocol. In this comparison, the more informative signal is WikiText-2, where the gap is much clearer (16.55 for naive prompting, 7.17 for step-by-step CoT, and 6.28 for GCoT), consistently showing the benefit of structured graph-based reasoning over simpler prompting variants.
>
> We agree that the capability of the search backbone may affect how much benefit GCoT can realize. To further validate this, we will add a small ablation using a stronger search backbone (GPT-5.4) while keeping the target model and other settings fixed:
>
> |Method|Step-by-step CoT|GCoT|MMLU|Wiki|
> |-|-:|-:|-:|-:|
> |Dense|✗|✗|40.52|5.12|
> |Naive|✗|✗|28.8|13.2|
> |step-by-step|✓|✗|31.0|6.55|
> |AutoPrune|✓|✓|32.1|5.88|
>
> We are grateful for your careful reading and valuable feedback, and we hope the above clarifications and added results help resolve your concerns.

---

> > ### Author Rebuttal · Reviewer_zKxG · 2026-04-03
> >
> > Appreciate authors response. I increase my score to 4.

---

> > > ### Author Response · Authors · 2026-04-04
> > >
> > > Dear Reviewer zKxG,
> > >
> > > Thank you very much for your thoughtful review, constructive comments, and encouraging feedback on our work. We are pleased to know that our rebuttal has addressed your concerns. We will carefully incorporate your suggestions into the final version of the paper. If you have any further questions or concerns, we would be happy to address them.
> > >
> > > Best Regards,
> > >
> > > Authors

---

### Decision · Program_Chairs · 2026-04-30

**Decision:**

Accept (regular)

**Comment:**

This paper introduces an interesting and timely perspective on LLM pruning by using an LLM-guided, graph-structured reasoning process to generate candidate pruning rules, and it pairs that idea with SDSA, a skewness-based layerwise sparsity allocation method aimed at mitigating failures at high sparsity. The main concerns in review were that the original framing overstated “self-discovery” and “no expert knowledge,” that important reproducibility and search-cost details were initially missing, and that the empirical gains over strong baselines such as Wanda were sometimes modest and mixed across zero-shot tasks. After reading the manuscript, rebuttal, and follow-up reviewer comments carefully, and the rebuttal substaintially improved the case on these points by clarifying the GCoT backbone and cost, explaining that the search is a one-time offline procedure transferred across settings, showing that the discovered rule helps even without SDSA, and adding evidence on Wanda-style grid search, cross-corpus selection, and search stability. There are still remaining limitations, i.e., the paper should describe its claims with more restraint, and the practical case is strongest for perplexity preservation and higher-sparsity regimes rather than for uniform dominance on downstream tasks. Even so, the core technical contribution is sound, the methodology is interesting, SDSA appears genuinely useful, and the empirical evidence is strong enough that the paper makes a worthwhile contribution to the efficient-LLM and model-compression literature. I therefore recommend acceptance.